# Italian Validation of the Short Version of the Failure to Mentalize Trauma Questionnaire in Adults at Risk Due to Childhood Trauma

**DOI:** 10.3390/bs13100843

**Published:** 2023-10-15

**Authors:** Giulia Raimondi, Claudio Imperatori, Sara Gostoli, Paola Gremigni, Marco Innamorati

**Affiliations:** 1Department of Human Sciences, European University of Rome, 00163 Rome, Italy; giulia.raimondi@unier.it (G.R.); claudio.imperatori@unier.it (C.I.); 2Department of Psychology “Renzo Canestrari”, University of Bologna, 40127 Bologna, Italy; sara.gostoli2@unibo.it (S.G.); paola.gremigni2@unibo.it (P.G.)

**Keywords:** adverse childhood experiences, mentalization, factor analysis, psychopathology, discriminant analysis

## Abstract

The impact of recurrent traumatic experiences during childhood may impede the integration of mentalization abilities and lead to psychopathology. Recently, the Failure to Mentalize Trauma Questionnaire (FMTQ), a comprehensive 29-item self-report scale aimed at identifying deficits in mentalization arising from childhood trauma, was developed. However, the length of the FMTQ may render it impractical for epidemiological studies involving multiple variables and measures. Furthermore, the initial testing revealed inadequate factor reliabilities for the two first-order factors. Therefore, this study aimed to shorten and create a unidimensional version (FMTQ-s) and investigate its psychometric properties, including internal consistency and convergent and concurrent validity, in a non-clinical Italian adult sample. The factor analysis supported a 13-item unidimensional version of the FMTQ with acceptable internal consistency (ordinal alpha = 0.88) and satisfactory convergent and concurrent validity. The FMTQ-s obtained scalar invariance between individuals with and without self-reported childhood traumas. Overall, the FMTQ-s appears to be a feasible and reliable tool for assessing deficits in mentalization resulting from childhood trauma.

## 1. Introduction

Childhood traumatic experiences, or adverse childhood experiences (ACEs), that involve repetitive emotional and/or physical neglect, emotional and/or physical abuse, and sexual abuse have been proven to be a form of developmental trauma (DT) that may result in serious psychopathology [1]. Such psychopathology includes trauma- and stressor-related disorders [2], dissociative disorders [3], personality disorders [4], eating disorders [5], and somatic symptoms and related disorders [6]. The attachment theory [7] provides a theoretical framework for ACEs. It highlights the importance of the caregiver–child relationship in shaping the individual’s emotional and social development and emphasizes how a disorganized attachment can have severe negative consequences in all life stages. It can lead to struggles with trust, self-esteem, and developing healthy relationships [8]. Moreover, studies have also demonstrated that the post-traumatic stress resulting from ACEs can also have an impact on brain development [9], making it difficult for individuals to regulate their emotions and deal with stress as they grow older.

Indeed, ACEs can lead to severe forms of psychopathology by disrupting the integration of mental functions [10,11] and negatively affecting one’s meta-cognition and mentalization abilities. Mentalization, or reflective functioning, refers to a psychological process that helps individuals to integrate their own and others’ mental states and regulate emotions [12]. There is a substantial body of research [13,14,15] assessing the mediating role of mentalization between childhood trauma and future psychopathology. A preserved ability to mentalize plays a critical role in individuals with traumatic experiences, as it can help them make sense of their traumas and integrate adverse experiences as part of their thought narrative [16], and could, therefore, represent a protective factor and promote resilience in survivors of childhood maltreatment [17].

However, it is important to note that mentalization is a context-specific ability, which means that reflective functioning might not be compromised by minor stressful situations, whilst it can be affected by trauma-specific situations [16]. For this reason, Berthelot et al. [18] have introduced the Failure to Mentalize Trauma Questionnaire (FMTQ), a self-report measure designed to identify mentalization deficits that may arise from traumatic experiences. Based on factor analysis, the FMTQ revealed a second-order structure (i.e., first-order factors all loading on a single higher-order factor) where all items loaded on seven dimensions of specific deficits in mentalization (i.e., Disorganization of Thoughts, Grandiosity, Absorption, Identification with the Victim, Identification with the Perpetrator, Avoidance of Thoughts, and Justification of Trauma) [18]. While the FMTQ has shown promising psychometric characteristics, such as construct and concurrent and predictive validity, its 29-item length presents a potential obstacle to its utilization in epidemiological or clinical studies. Shorter questionnaires are often preferable, as they permit the administration of multiple measures and alleviate the burden on patients. Additionally, two first-order factors (i.e., Identification with the Victim and Justification of Trauma) exhibit less than optimal psychometric properties (Cronbach’s alphas of 0.62 and 0.60, respectively) [17], indicating the need for refinement. Therefore, this current study aims to refine a short version of the FMTQ (FMTQ-s) and examine its psychometric properties (i.e., internal consistency and convergent and concurrent validity) in a non-clinical population of Italian adults. The goal is to create a unidimensional short version of the FMTQ that takes into account the presence of a higher-order general factor detected in the original study. We chose to validate the questionnaire on a non-clinical population because adverse childhood experiences are also common in non-clinical individuals, and thus, an instrument assessing mentalization deficits due to past traumas also validated on a sample from the general population could be helpful for epidemiological studies.

## 2. Method

### 2.1. Participants

A group of 709 Italian adults, comprising 637 women and 72 men, with a mean age of 27.24 years (SD = 8.00), were recruited as a convenience sample through social networks such as Facebook, LinkedIn, and Instagram, addressing open groups, from September 2021 to May 2022 (see Table 1 for socio-demographic details of the sample). The participants were required to fulfill certain inclusion criteria, like an age of 18 years and older, fluency in Italian language to complete the battery of tests, and the provision of informed consent. It is notable that the participants agreed to participate in the study voluntarily and did not receive any form of payment or compensation, such as academic credits. The study was conducted following the ethical guidelines set forth by the Helsinki declaration standards and was approved by the Ethics Committee of the European University of Rome (protocol no. 05/2021).

### 2.2. Measures

The participants in the study underwent an evaluation that assessed a variety of socio-demographic and clinical variables, including age, sex, educational attainment, marital status, job status, use of tobacco and/or alcohol, and legal highs in the past six months. Additionally, they completed several questionnaires that measured different aspects of mental health. These included the Failure to Mentalize Questionnaire (FMTQ) [18], the Mentalization Questionnaire (MZQ) [19,20], the short form of the Childhood Trauma Questionnaire (CTQ) [21,22], and the Symptoms-Checklist-K-9 (SCL-K-9) [23,24].

The FMTQ is a self-report measure that assesses different types of mentalization failure due to past traumas and consists of 29 items organized into the following 7 dimensions: (1) Disorganization of Thought refers to difficulties in reasoning during the experience of dissociative symptoms due to trauma-related emotions; (2) Grandiosity reflects one’s own belief to be immune to the experience of traumas; (3) Absorption refers to the experience of intrusive memories related to traumatic situations; (4) Identification with the Victim assesses an individual’s belief that they are to blame for every past traumatic experience; (5) Identification with the Perpetrator evaluates an individual’s method of dealing with negative emotions through aggressive behaviors; (6) Avoidance of Thoughts measures the voluntary choice of avoiding thinking about difficult memories and feelings related to past traumas; and (7) Justification of Trauma assesses an individual’s belief that other people’s mean behaviors need to be justified and are actually well intended. A global score is also computed, obtained by summing the scores of all dimensions. Items are rated on a 5-point Likert scale from 0 = “*Strongly disagree*” to 4 = “*Strongly agree*”, with higher scores indicating greater failures in the mentalization of trauma(s) and adverse relationships for each specific dimension, as well as for the global score. The Italian version of the FMTQ was obtained using a well-established back-translation procedure to ensure accurate translation. First, one author of the study (G.R.) forward-translated the FMTQ from English to Italian. Second, another author of the study (C.I.) back-translated the Italian version of the FMTQ into English, blind to the original English version. Finally, a third author (M.I.) checked for the presence of errors and ambiguities.

The MZQ [19] is a self-report questionnaire consisting of 13 items intended to measure the lack of mentalization abilities. The Italian validation of the MZQ [20] confirmed a one-factor solution with 13 items, which deviates from the original version’s four-factor solution with 15 items [19]. A low score reflects a high deficit in mentalization, and items are rated on a 5-point Likert scale ranging from 1 = “*I agree*” to 5 = “*I disagree*”. Cronbach’s alpha for the current study was 0.85.

The CTQ-SF [21,22] is a 28-item self-report questionnaire containing 28 items that assess multiple forms of childhood trauma, namely Emotional Abuse (EA), Emotional Neglect (EN), Physical Abuse (PA), Physical Neglect (PN), and Sexual Abuse (SA). The items are rated using a 5-point Likert scale from 1 = “*Never true*” to 5 = “*Very often true*”, with higher scores reflecting more severe traumatic experiences during childhood. Cronbach’s alphas for the current study were EA = 0.87, EN = 0.90, PA = 0.88, PN = 0.65, and SA = 0.93.

Lastly, the SCL-K-9 [23,24] is the 9-item shortened version of the Symptoms Checklist-90-Revised (SCL-90-R) [25]. The SCL-K-9 consists of 9 items and measures overall psychological distress on a 5-point Likert scale ranging from 0 = “*None at all*” to 4 = “*Very severe*”. The total score of the SCL-K-9 is commonly recorded as the Global Severity Index (GSI-K-9). Cronbach’s alpha for the current study was 0.82.

### 2.3. Statistical Analysis

A unidimensional abbreviated version of the questionnaire (FMTQ-s) was developed through an exploratory approach. The sample was randomly split into two subsamples using the SPSS function. In the first sample, an exploratory factor analysis (EFA) was performed, and items with factor loadings less than 0.40 were removed. Additionally, large modification indices (M.I. > 10) were inspected in order to make adjustments to improve the model. In the second subsample, a confirmatory factor analysis (CFA) was performed to confirm the stability of the factor solution obtained in the previous step. The number of participants required for a CFA is strongly affected by many variables: the number of latent dimensions, the number of factor loadings, the number of covariances among latent dimensions, the data scaling (i.e., categorical versus continuous), the estimator type (e.g., WLSMV, ML, robust ML, etc.), and the missing data and model complexity [26,27]. Therefore, for the CFA, a number of individuals ranging between 200 and 500 participants is usually recommended [26,27]. The models from both EFA and CFA were evaluated using the Mean- and Variance-Adjusted Weighted Least Square (WLSMV) estimator with a polychoric correlation matrix. The fitness of the models for both EFA and CFA was assessed using absolute and incremental fit indices, namely the chi-square (χ^2^) test, the Root Mean Square Error of Approximation (RMSEA), the Standardized Root Mean Square Residual (SRMR), the Comparative Fit Index (CFI), and the Tucker–Lewis Index (TLI). Given that the chi-square fit statistic is affected by large sample size, the ratio of the chi-square statistic to the respective degrees of freedom (χ^2^/df) is preferred [28]. A ratio of ≤ 5 indicates a superior fit between the hypothesized model and the sample data [29]. For RMSEA, values below 0.05 indicate a good fit, values between 0.05 and 0.08 indicate an acceptable fit, and values equal to or greater than 0.10 indicate a poor fit [30]. TLI and CFI values of 0.95 and higher indicate good model fit, while values of 0.90 and higher indicate an acceptable fit. SRMR values below 0.08 indicate good fit [30].

A multigroup CFA was subsequently performed to evaluate the measurement invariance of the refined one-factor model among individuals with and without self-reported childhood traumas [31]. The measurement invariance was tested with a three-step procedure. Firstly, configural invariance was tested to determine if the same number of factors is extracted across groups [32]. Secondly, metric invariance was tested to evaluate if the same items load on the same factor across groups [32]. The achievement of metric invariance signals that the factor loadings of a particular construct are indicative of equivalent responses from individuals belonging to diverse groups. This connotes a shared understanding of the items among the participants, establishing the uniformity of the underlying factor’s significance across dissimilar groups. Thirdly, scalar invariance was tested to assess if items’ intercepts (i.e., thresholds) are equivalent across groups. Scalar invariance ensures that the items not only have the same metrics but also the same origin and belong to the same scale. When scalar invariance is achieved, it indicates that individuals from different groups who possess identical levels of the latent variable would demonstrate equal scores on the corresponding items. Any variations observed in the mean values of the respective items can be ascribed to divergences in the participants’ levels of the latent factor. Consequently, scalar invariance facilitates the comparison of mean differences of the latent factor between groups [32]. The study used the same fit indices to assess the three invariance models. The two groups (childhood trauma vs. no childhood trauma) were classified based on their childhood trauma history as reported in the CTQ subscales. The childhood trauma group comprised individuals who reported experiencing at least one CTQ trauma category (i.e., Emotional Abuse, EA; Emotional Neglect, EN; Physical Abuse, PA; Physical Neglect, PN; and/or Sexual Abuse, SA). The cut-off used to determine the presence of self-reported childhood trauma was the same as that used by Berthelot et al. [18].

The present study computed indices of internal consistency, namely ordinal alpha, the Molenaar–Sijtsma statistic (MS), and the latent class reliability coefficient (LCRC), across the entire sample. A Type III ANOVA model was used to assess the presence of any sex differences. Correlations with other measures were calculated using Pearson r correlation coefficients, with values of less than 0.10 indicating negligible or null correlation, values between 0.10 and 0.30 small correlation, values between 0.30 and 0.50 medium correlation, and values exceeding or equal to 0.50 large correlation, according to Cohen [33]. Furthermore, we investigated discriminant validity between the FMTQ-s and the MZQ following the approach recommended by Meng [34] to further establish the efficacy of the FMTQ-s. Specifically, we examined whether correlations between the FMTQ-s scores and factors of CTQ-SF were significantly greater than the corresponding correlation with the MZQ. The results were presented in the form of 95% Confidence Intervals (CI) and *p*-values, with a significant *p*-value set at <0.05.

All analyses were performed using Mplus 8.3 [35], version 3.0.4 of the Mokken package in R [36], and SPSS 25 [37].

## 3. Results

### 3.1. Factor Structure of the Italian FMTQ-s

In the first subsample (N = 378), the refinement process of the one-factor model indicated 16 items with low factor loadings (<0.40). Then, M.I.s of the remaining items were inspected, which suggested the need to include paths between the errors of two pairs of items (items #12 and #26; items #18 and #23). The CFA on the second subsample (N = 331) of the FMTQ-s with 13 items indicated an adequate fit (χ^2^(63) = 241.87; χ^2^/DF = 3.84; RMSEA = 0.08, 90% CI = 0.07–0.09; CFI = 0.94; TLI = 0.93; SRMR = 0.05) (see Figure 1 for standardized factor loadings) (see the Appendix A for the items excluded).

### 3.2. Psychometric Properties of the Italian FMTQ-s

To test the measurement invariance, the sample was divided into two groups of individuals with (N = 336) and without (N = 373) childhood trauma. The two groups did not differ in sex (χ^2^ = 0.28, *p* = 0.60) but reported significantly different levels of general psychopathology (GSI-K-9) (t_4.67_ = 11.42, *p* = 0.03). Individuals with self-reported childhood trauma reported greater levels of general psychopathology (M = 19.84, SD = 6.50) compared to other respondents (M = 14.10, SD = 6.94). Scalar invariance was obtained for the FMTQ-s (χ^2^ = 671.15 (266); χ^2^/ DF = 2.52; RMSEA = 0.066, 95% CI: 0.060–0.072; CFI = 0.93; TLI = 0.94; SRMR = 0.06) (Table 2).

The FMTQ-s showed satisfactory internal consistency (ordinal alpha = 0.88, MS = 0.85, and LCRC = 0.86). No relevant floor or ceiling effects were evident. The average score was 21.30 (SD = 10.13) (ranging from 0 to 52, with a higher score reflecting greater mentalization deficits due to past trauma). The ANOVA results did not indicate sex differences (F_1-729_ = 2.87, *p* = 0.09) for the mean score of the FMTQ-s (women: mean score = 19.02, SD = 9.29; men: mean score = 17.11, SD = 8.17). Correlations among variables are reported in Table 3. Specifically, the FMTQ-s reported greater correlation with all dimensions of the CTQ compared to the MZQ, except for Sexual Abuse (r = 0.11, *p* < 0.01). Correlations with age were also computed; however, the effect sizes were all negligible with all of the outcome measures, and two non-significant correlations with CTQ Physical Abuse (r = 0.06, *p* = 0.10) and Physical Neglect (r = 0.04, *p* = 0.27) were found.

The correlations with the FMTQ-s were significantly greater than the correlations with the MZQ. Furthermore, there was no significant difference between the correlations of the Sexual Abuse subscale with the FMTQ-s and with the MZQ (r difference = −0.01, *p* = 0.88) (see Table 4).

## 4. Discussion

In light of the constraints posed by the length and suboptimal internal consistency of the two first-order dimensions of the FMTQ, as highlighted by Berthelot et al. [18], we have devised a 13-item unidimensional version of the questionnaire, known as the FMTQ-s.

In our analyses, we found that the fit indices were satisfactory, along with good internal consistency and congruent and convergent validity. However, upon inspecting the M.I.s, it was suggested to add a correlation between the errors of two pairs of items (item #12 with #26, both referring to difficulties in talking about past traumatic experiences; item #18 with #23, both referring to the act of brooding over the past traumatic experiences) [18]. The decision to include correlations between those pairs of items was also based on the fact that those items are similar in both their written form and their meaning.

The factor structure of the FMTQ-s has been shown to remain invariant across individuals with different experiences of childhood trauma. This indicates that not only do both groups interpret the items in a similar manner and that the factor loadings remain consistent across groups, but also that the factor mean scores can be compared, as the scale’s origin is the same for both groups. The results of our study revealed that the individuals who report experiencing childhood trauma have greater mentalization deficits compared to those who do not report such experiences due to the trauma to which they are exposed. These findings corroborate previous research that also reported greater mentalization deficits in people who experienced childhood trauma [15,38,39]. Although we controlled for sex differences between the two groups, the proportion of female respondents was higher. As recently indicated by Becker [40], there is still a lack of research regarding this common phenomenon affecting surveys world-wide. Becker provided several theoretical hypotheses to explain why women are more prone than men to take part in online surveys, such as gender differences regarding helping, social influence, and perception of costs and benefits. It could also be due to the fact that women seem to prefer to complete surveys on smartphones, whilst men seem to prefer to complete surveys on computers, which makes them less easy to reach [40]. However, since there is still not enough research on this topic, we cannot establish with certainty the reasons for such a high disparity between women and men responding to our survey.

Regarding the unbalanced proportion of women and men in our sample, in the Type III model used in ANOVA, sums of squares are invariant with respect to the cell frequencies. This type of sums of squares is often considered useful for an unbalanced model with no missing cells. Moreover, the assumption of equal variances across groups was not violated. Thus, it seems that ANOVA results can be considered reliable, despite the unequal sample size of the groups.

The congruent and convergent validity of the FMTQ-s was found to be strong. This was evidenced by its positive and substantial correlation with a measure of general psychopathology. This aligns with previous research that has suggested a relationship between deficits in mentalization and traumatic experiences across various mental health conditions. In particular, it has been established that childhood trauma is strongly associated with increased mentalization deficits that may contribute to later susceptibility to mental health problems, including depression, anxiety, dissociation, and self-harm [41,42,43,44].

Finally, our results also highlighted significant correlations between FMTQ-s scores and all the CTQ dimensions and demonstrated discriminant validity with the MZQ. Notably, the FMTQ-s showed significantly stronger correlations with these measures than did the MZQ, except for the correlation with Sexual Abuse. These findings not only offer support for the construct validity of the shortened FMTQ-s, which measures specific aspects of mentalization tied to traumatic experiences differently from the MZQ, but they also provide support for the fact that mentalization is a context-specific ability and that the FMTQ-s does not measure mentalization in general but specific failures in how one mentalizes about trauma and adverse relationships. However, it is important to highlight that although the differences among the correlations between the FMTQ-s and the MZQ are significant, indicating the discriminant validity of the FMTQ-s, the bivariate correlations are relatively small. This could indicate that, from a clinical perspective, both instruments might not be sufficiently powerful to explain the variability of mentalization. Moreover, this could be due to the non-clinical nature of the sample or to the fact that other clinical variables associated with the construct of mentalization need to be considered.

### Limitations and Future Directions

Several limitations must be kept in mind when interpreting the results of the current study. Firstly, our sample was skewed toward women, so our findings may not be generalizable to the broader population. Secondly, our research relied on self-reported information about participants’ childhood trauma, rather than on clinical interviews. Therefore, future studies should verify the validity of the FMTQ-s in individuals from clinical populations. Thirdly, self-report measures can be affected by multiple response biases, such as social desirability [45]. Fourthly, it is important to note that the full-length FMTQ [18] is not validated in Italian yet; therefore, future studies should compare the original version and this shortened version of the FMTQ in order to assess their psychometric properties. Fifthly, the CFA model was adjusted by correlating the error variances of two pairs of items, which is a data-driven approach. Finally, our study was cross-sectional, so we cannot affirm the stability of the FMTQ-s (i.e., test–retest validity) and measurement invariance over time. Future studies addressing these concerns will allow researchers to fully assess the potential of the FMTQ-s as a valid and reliable tool for assessing the mentalization of individuals who have experienced trauma.

## 5. Conclusions

In summary, the present study’s findings suggest that the FMTQ-s could serve as a highly viable, valid, and reliable instrument for assessing deficits in mentalization related to traumatic experiences. The study indicates that the abbreviated version of the FMTQ could be particularly advantageous for screening and evaluating populations, as it is less time consuming than the original FMTQ. Nonetheless, researchers must conduct further research to fully evaluate the psychometric properties of the FMTQ-s in clinical populations. Such studies could yield valuable insights into the fundamental concept of mentalization, enabling researchers and clinicians to devise more individualized interventions. As such, there is a clear need for continued investigation into how the FMTQ-s can enhance our understanding and assessment of mentalization-related deficiencies in individuals who have experienced trauma.

## Figures and Tables

**Figure 1 behavsci-13-00843-f001:**
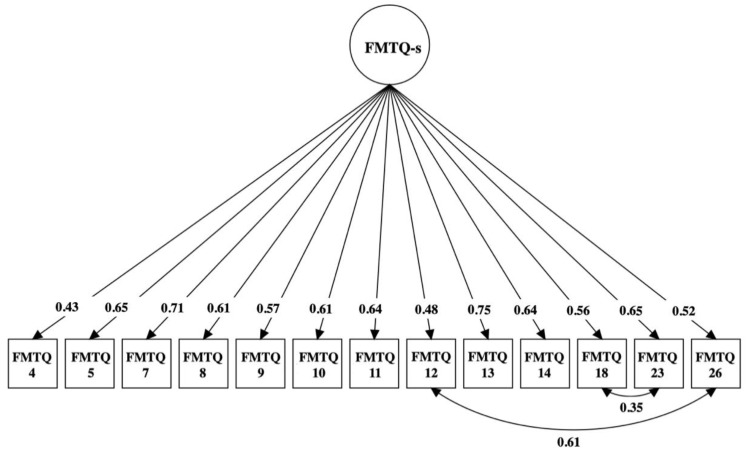
Standardized loadings of the FMTQ-s.

**Table 1 behavsci-13-00843-t001:** Descriptive statics of the sample (*N* = 709).

*Variables*	*N|M*	*%|(SD)*
**Age** M(SD)	27.24	(8.00)
**Sex** N/%		
Men	72	10.2%
Women	637	89.8%
**Job status** N/%		
Employed	304	42.9%
Unemployed or retired	98	13.8%
Student	307	43.3%
**Marital status** N/%		
Single	489	69%
Married/stable relationship	202	28.5%
Divorced/widowed	18	2.5%
**Use of tobacco** N/%	314	44.3%
**Alcohol** N/%	456	64.3%
**Use of illegal drugs** N/%	67	9.4%

***Note.*** *N* = number; *M* = mean; *%* = percentage; *SD* = standard deviation.

**Table 2 behavsci-13-00843-t002:** Measurement of invariance between people with and without childhood trauma.

Model	χ^2^ (df)	RMSEA (90% CI)	CFI	TLI	SRMR	Model Comparison	χ^2^ (Δdf)	*p*	Results
**Configural**	463.06 (126)	0.088 (0.079–0.096)	0.94	0.92	0.05	-	-	-	Accepted
**Metric**	453.63 (138)	0.081 (0.073–0.089)	0.95	0.94	0.05	**Configural**	6.92 (12)	0.86	Accepted
**Scalar**	420.93 (176)	0.063 (0.056–0.071)	0.96	0.96	0.05	**Metric**	35.08 (38)	0.60	Accepted

*N* = 709: trauma group *N* = 336; no trauma group *N* = 373.

**Table 3 behavsci-13-00843-t003:** Correlations among variables (N = 709).

	FMTQ-s	MZQ	GSI-K-9	CTQ_PA	CTQ_SA	CTQ_EA	CTQ_PN
**MZQ**	−0.68 **						
**GSI-K-9**	0.61 **	−0.65 **					
**CTQ_PA**	0.26 **	−0.22 **	0.23 **				
**CTQ_SA**	0.10 **	−0.11 **	0.16 **	0.22 **			
**CTQ_EA**	0.42 **	−0.38 **	0.44 **	0.61 **	0.23 **		
**CTQ_PN**	0.26 **	−0.20 **	0.26 **	0.57 **	0.26 **	0.57 **	
**CTQ_EN**	0.37 **	−0.35 **	0.37 **	0.51 **	0.18 **	0.74 **	0.65 **

***Note.*** **, All *p* < 0.001. FMTQ-s = Failure to Mentalize Trauma Questionnaire—short version; MZQ = Mentalization Questionnaire; GSI-K-9 = Global Severity Index of the Symptoms Checklist-90-Revised; CTQ_PA = Physical Abuse subscale of the Childhood Trauma Questionnaire; CTQ_SA = Sexual Abuse subscale of the Childhood Trauma Questionnaire; CTQ_EA = Emotional Abuse subscale of the Childhood Trauma Questionnaire; CTQ_ON = Physical Neglect subscale of the Childhood Trauma Questionnaire; CTQ_EN = Emotional Neglect subscale of the Childhood Trauma Questionnaire.

**Table 4 behavsci-13-00843-t004:** Differences between correlation of the FMTQ-s and MZQ with the CTQ subscales.

	r_diff	95% CI	*p*
**CTQ_PA**	0.48	0.34–0.55	<0.001
**CTQ_SA**	−0.01	−0.14–0.12	0.88
**CTQ_EA**	0.8	0.61–0.75	<0.001
**CTQ_PN**	0.46	0.32–0.54	<0.001
**CTQ_EN**	0.72	0.54–0.71	<0.001

***Note.*** r_diff = differences between correlations; 95% CI = 95% Confidence Intervals; CTQ_PA = Physical Abuse subscale of the Childhood Trauma Questionnaire; CTQ_SA = Sexual Abuse subscale of the Childhood Trauma Questionnaire; CTQ_EA = Emotional Abuse subscale of the Childhood Trauma Questionnaire; CTQ_ON = Physical Neglect subscale of the Childhood Trauma Questionnaire; CTQ_EN = Emotional Neglect subscale of the Childhood Trauma Questionnaire.

## Data Availability

The data that support the findings of this study are available from the corresponding author, G.R., upon reasonable request.

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
