# Peer review of "Italian Validation of the Short Version of the Failure to Mentalize Trauma Questionnaire in Adults at Risk Due to Childhood Trauma"

_behavsci, 2023, doi:10.3390/bs13100843_

Round 1

Reviewer 1 Report

The study proposes an Italian validation of an instrument to assess deficits in mentalization akin to childhood trauma. This could be a relevant contribution because it is necessary to have good and reliable tools that are also agile in terms of the number of items.

Although the authors' proposal is interesting, I believe the paper suffers from some limitations that should be highlighted more emphatically.

The main problem I see is the unbalanced sample.

Although the number of males and females is highly unbalanced, they were considered as a unique sample. Can the authors justify this difference in recruitment? I wonder if the authors can make some points about the large difference in numbers, even if they controlled for the sex-related effect. The authors report this as a limitation to the study, but in my opinion, this should be explored further in the discussion because it limits the generalizability of the results.

Even with respect to internal consistency, which appears moderate, can the authors try to better justify the result, or at least fit it within the limitations of the study?

In the “Introduction” section, It sounds strange to find for the definition of the concept of mentalization a theoretical reference from 2014.... (ref. no. 9). I think it is more appropriate to refer to more recognized authors in the literature landscape for definitions of such a fundamental concept. There is also a lack of theoretical references to the concept of Adverse Childhood Experiences (ACEs).

Finally, some minor remarks:

-in the title, I suggest changing the term “individuals” to “adults”.

- in the affiliation, there are letters next to names and numbers next to Departments.

Author Response

The study proposes an Italian validation of an instrument to assess deficits in mentalization akin to childhood trauma. This could be a relevant contribution because it is necessary to have good and reliable tools that are also agile in terms of the number of items. Although the authors' proposal is interesting, I believe the paper suffers from some limitations that should be highlighted more emphatically.

Authors’ answer: We would like to thank the Reviewer for all the helpful comments on the manuscript.

The main problem I see is the unbalanced sample. Although the number of males and females is highly unbalanced, they were considered as a unique sample. Can the authors justify this difference in recruitment? I wonder if the authors can make some points about the large difference in numbers, even if they controlled for the sex-related effect.The authors report this as a limitation to the study, but in my opinion, this should be explored further in the discussion because it limits the generalizability of the results.

Authors’ answer: thank you for pointing this out. We have better explored this topic in the discussion section, as you suggested.

Even with respect to internal consistency, which appears moderate, can the authors try to better justify the result, or at least fit it within the limitations of the study?

Authors answer: Although there is not a consensus among researchers about the optimal value of Cronbach' alpha, we considered an alpha of 0.88 very good by taking into consideration that if alpha is too high it may suggest that some items are redundant, therefore a maximum alpha value of 0.90 has been recommended [Strainer, D. L. (2003). Starting at the beginning: an introduction to coefficient alpha and internal consistency. Journal of Personality Assessment, 80:99-103. DOI: 10.1207/S15327752JPA8001_18]. Moreover, Nunnally and Bernstein [Nunnally, J. C., & Bernstein, I. H. (1994). Psychometric Theory. New York. NY: McGraw-Hill.] suggested that for correlational study, a reliability ≥ 0.80 is to be considered appropriate.

In the “Introduction” section, It sounds strange to find for the definition of the concept of mentalization a theoretical reference from 2014.... (ref. no. 9). I think it is more appropriate to refer to more recognized authors in the literature landscape for definitions of such a fundamental concept. There is also a lack of theoretical references to the concept of Adverse Childhood Experiences (ACEs).

Authors’ answer: We apologize for the mistake made in the reference number’s order; the correct reference  # 9 was the article of Fonagy et al., 2016, not that of 2014. We have corrected all the references list order; however, since new references were added, the former reference # 9 (Fonagy et al., 2016) is now reference # 12. We have also added a wider theoretical background to the concept of ACEs in the introduction.

Finally, some minor remarks:

-in the title, I suggest changing the term “individuals” to “adults”.

- in the affiliation, there are letters next to names and numbers next to Departments.

 Authors’ answer: The title has now been changed as suggested. The letters next to the names have now been changed into numbers.

Reviewer 2 Report

Thank you for the opportunity to review this interesting study. I will begin with a short summary of how I interpret the paper, followed by specific comments.

This study is a validation of the FMTQ, which shows that a 13-item short form of the Italian version of this study shows satisfactory fit, good internal consistency, and good congruent and convergent validity. The authors conclude that the short-form FMTQ-s can be used as a valid instrument for assessing mentalization of childhood trauma.

Firstly, I would like to congratulate the authors on a well-designed and thought-out study to validate a measure, which is always very appreciated. Below, I outline a few questions and concerns, but overall, I think the quality is very high in this manuscript. My main concern is regarding the possibility of over-analyzing the statistically significant correlations between measurement tools. Some of the correlations are quite small, and I wonder about the clinical relevance of these studies. I do, however, understand that this is a non-clinical population, and appreciate the authors discussion of the need for future research on a clinical sample.

Introduction:

 1.    Line 61-62: Why have the authors chosen to use a non-clinical population? Is the questionnaire sensitive enough to be tested in this population (that is, compared to survivors of childhood maltreatment)? Would it be more relevant to have tested on a clinical population?

Methods:

2.       Line 68: Was any sample-size calculation performed, or was the aim to collect as much data as possible? It may also be relevant to describe the recruitment procedure more thoroughly, for example using closed or open groups, friends, shares, how long was the data collection period?

3.       Line 69: “convenient sample” should read “convenience sample”.

4.       Line 72: It appears the authors have missed a word. I assume it is meant to read “an age of 18 years…”.

5.       Line 78 (Table 1): Is the use of “Sex” as opposed to “Gender” a conscious choice?

6.       Line 78 (Table 1): I think it would be more easily understood if “Smoking” were changed to “Smoker”.

a.       On line 99 the authors describe it as “use of tobacco”, so this might be a better term to use in the table to maintain consistency and not confuse between smoking and other forms of tobacco use?

7.       Line 78 (Table 1): I am unsure of what “Use of illegal highs” means – is this perhaps the use of illegal drugs, for example?

8.       Lines 101-102: The reference for the FMTQ does not seem to follow the same format as other references (i.e. it is displayed as author, year).

9.       Lines 101-104: Did the authors use validated versions of these questions in Italian? The reference provided for the FMTQ, for example, is validated in French.

a.       I see now that this is explained starting on line 120.

10.   Lines 118-120: It might be beneficial to add a total score to make it easier for the reader to interpret the final scores reported in the results (i.e. are the results reported on lines 230-238 and indication of positive or negative mentalization?).

11.   Lines 169-170: The authors here report that an analysis was done based on a grouping of exposure to childhood traumas (or lack thereof). However, on line 61-62 the population is described as non-clinical. Is this grouping large enough and/or sensitive enough to detect differences between these groups?

Results

12.   Lines 232-233: Is it relevant also to provide demographic information, i.e. regarding age, in this result?

13.   Line 252 (Table 3): These results, although statistically significant, show in some cases quite small correlation coefficients.

Discussion

14.   Line 282: For correct (and slightly picky!) English, this should read “… the trauma to which they have been exposed, compared…”.

15.   Line 293: The authors indicate “associations” and seem to use this term interchangeably with “correlations.” I would recommend consistency in choice of terminology.

16.   293-301: See comment 12 (Results section regarding statistical significance of correlations). It might be worth discussing this aspect more thoroughly to be transparent with the fact that these are in fact relatively small differences. This would be especially helpful if there are clinically relevant differences which may highlight what could be considered relevant in these measures as opposed to simply looking at weak to moderate correlations. That is, does the statistical significance provide any information which would indicate that this is still a measure which captures the constructs in question?

17.       Lines 302-313: I believe it may be relevant to mention that the FMTQ (the original version) is not validated in Italian. Despite having translated according to established guidelines, there are procedures which can ensure that the translation is correct. 

18.   Lines 304-305: This sample seems to be very skewed towards women. Is there any reason for this? Perhaps the sampling method was skewed towards Facebook groups targeting female survivors or childhood maltreatment as opposed to males? Based on the analysis of psychometric properties, it appears that sex differences were examined (Line 233), but no significant differences were found, which seems surprising given the sheer number of females compared to males.

I have very few concerns about the use of English in this manuscript. Those instances where I believe improvement can be made are included in my comments to the authors.

Author Response

Thank you for the opportunity to review this interesting study. I will begin with a short summary of how I interpret the paper, followed by specific comments.

This study is a validation of the FMTQ, which shows that a 13-item short form of the Italian version of this study shows satisfactory fit, good internal consistency, and good congruent and convergent validity. The authors conclude that the short-form FMTQ-s can be used as a valid instrument for assessing mentalization of childhood trauma. Firstly, I would like to congratulate the authors on a well-designed and thought-out study to validate a measure, which is always very appreciated. Below, I outline a few questions and concerns, but overall, I think the quality is very high in this manuscript. My main concern is regarding the possibility of over-analyzing the statistically significant correlations between measurement tools. Some of the correlations are quite small, and I wonder about the clinical relevance of these studies. I do, however, understand that this is a non-clinical population, and appreciate the authors discussion of the need for future research on a clinical sample.

Authors’ answer: We would like to thank the Reviewer for all the helpful comments on the manuscript.

Introduction:

  1. Line 61-62: Why have the authors chosen to use a non-clinical population? Is the questionnaire sensitive enough to be tested in this population (that is, compared to survivors of childhood maltreatment)? Would it be more relevant to have tested on a clinical population?

Authors’ answer: We validated the questionnaire on a non-clinical population because adverse childhood experiences are common also in non-clinical individuals and, therefore, an instrument assessing mentalization deficits due to past traumas validated also on a sample from the general population could be helpful for epidemiological studies. We have added this consideration at the end of the Introduction. Moreover, the measurement of invariance analysis, allowed us to verify the adequacy of the factor structure across the two groups of individuals with and without self-reported childhood trauma (which is a construct often measured retrospectively and through self-reported questionnaires also in clinical populations). In the future directions section we better specified that this instrument is ought to be validated in individuals from different clinical populations.

Methods:

  1. Line 68: Was any sample-size calculation performed, or was the aim to collect as much data as possible? It may also be relevant to describe the recruitment procedure more thoroughly, for example using closed or open groups, friends, shares, how long was the data collection period?

Authors’ answer: Concerning the minimum sample sized, we have followed the indication reported in the literature regarding factor analysis, which suggest a minimum number of individuals to be ranged between 200 and 500. We thank the Reviewer for pointing this out and it was also specified in the statistical analyses section. We also explain better the recruited procedure in the Participants section: we used social networks addressing open groups and have added the period of the data collection.

  1. Line 69: “convenient sample” should read “convenience sample”.

Authors’ answer: The typo has been corrected.

  1. Line 72: It appears the authors have missed a word. I assume it is meant to read “an age of 18 years…”.

Authors’ answer: Yes, thank you. The typo has been corrected.

  1. Line 78 (Table 1): Is the use of “Sex” as opposed to “Gender” a conscious choice?

Authors’ answer: Yes it is, we wanted to clarify that the categories of “Female” and “Male” referred to the biological sex, not gender identity.

  1. Line 78 (Table 1): I think it would be more easily understood if “Smoking” were changed to “Smoker”.

Authors’ answer: The name was changed as “use of tobacco” has suggested in the next comment.

  1. On line 99 the authors describe it as “use of tobacco”, so this might be a better term to use in the table to maintain consistency and not confuse between smoking and other forms of tobacco use?

 Authors’ answer: We now have used the same description in the Table as well.

  1. Line 78 (Table 1): I am unsure of what “Use of illegal highs” means – is this perhaps the use of illegal drugs, for example?

 Authors’ answer: Yes it is. We have now changed the terms “illegal highs” into “illegal drugs” for clarity.

  1. Lines 101-102: The reference for the FMTQ does not seem to follow the same format as other references (i.e. it is displayed as author, year).

Authors’ answer: The reference in the text have been corrected as you suggested. Moreover, we have checked all the other references as well and corrected the number order of them all.

  1. Lines 101-104: Did the authors use validated versions of these questions in Italian? The reference provided for the FMTQ, for example, is validated in French.

I see now that this is explained starting on line 120.

Authors’ answer: Yes, we have described the procedure in the method section.

  1. Lines 118-120: It might be beneficial to add a total score to make it easier for the reader to interpret the final scores reported in the results (i.e. are the results reported on lines 230-238 and indication of positive or negative mentalization?).

Authors’ answer: A total score was added to the description of the questionnaire and the lines indicated in the results were better explained. Moreover, the range for total score of the FMTQ-s was provided in the “Psychometric properties of the Italian FMTQ-s” paragraph of the Results.

  1. Lines 169-170: The authors here report that an analysis was done based on a grouping of exposure to childhood traumas (or lack thereof). However, on line 61-62 the population is described as non-clinical. Is this grouping large enough and/or sensitive enough to detect differences between these groups?

Authors’ answer: Thank you for this comment. The CTQ is a well-validated tool for assessing childhood trauma in both clinical and non-clinical populations. Moreover, the cut-off used for the current study was the same used by the authors of the FMTQ, and in previous studies, https://jamanetwork.com/journals/jamapsychiatry/article-abstract/205109 . Finally, the sample sizes of the two groups were large enough to perform the measurement of invariance analysis.

Results

  1. Lines 232-233: Is it relevant also to provide demographic information, i.e. regarding age, in this result?

Authors’ answer: We have thought that providing information about gender and psychopathology was significant for two reasons: 1) the high unbalanced number of women and men in the sample was found, and 2) since there is an extensive literature supporting the fact that childhood trauma is associated to greater psychopathology symptoms in individuals from different clinical populations, we wanted to assess whether this difference was also true for non-clinical individuals. Moreover, correlation between age and the other construct were computed, however all correlations had negligible effect sizes, and there were also non-significant correlations with two dimensions of the CTQ (i.e., physical abuse and physical neglect). We have now added this result in the “Psychometric properties of the Italian FMTQ-s” paragraph of the Results.

  1. Line 252 (Table 3): These results, although statistically significant, show in some cases quite small correlation coefficients.

Authors’ answer: In light of the comment no.16, we have better highlighted this consideration in the discussion section.

Discussion

  1. Line 282: For correct (and slightly picky!) English, this should read “… the trauma to which they have been exposed, compared…”.

Authors’ answer: The typo has been corrected.

  1. Line 293: The authors indicate “associations” and seem to use this term interchangeably with “correlations.” I would recommend consistency in choice of terminology.

Authors’ answer: We changed the term “association” into “correlation” as you suggested.

  1. 293-301: See comment 12 (Results section regarding statistical significance of correlations). It might be worth discussing this aspect more thoroughly to be transparent with the fact that these are in fact relatively small differences. This would be especially helpful if there are clinically relevant differences which may highlight what could be considered relevant in these measures as opposed to simply looking at weak to moderate correlations. That is, does the statistical significance provide any information which would indicate that this is still a measure which captures the constructs in question?

Authors’ answer: We have better highlighted this consideration in the discussion section.

  1. Lines 302-313: I believe it may be relevant to mention that the FMTQ (the original version) is not validated in Italian. Despite having translated according to established guidelines, there are procedures which can ensure that the translation is correct.

Authors’ answer: Thank you for pointing this out. We have specified that the FMTQ (original version) was not previously validated in Italian in the limitations.

  1. Lines 304-305: This sample seems to be very skewed towards women. Is there any reason for this? Perhaps the sampling method was skewed towards Facebook groups targeting female survivors or childhood maltreatment as opposed to males? Based on the analysis of psychometric properties, it appears that sex differences were examined (Line 233), but no significant differences were found, which seems surprising given the sheer number of females compared to males.

Authors’ answer: We have better discussed the fact that the number of males and females is highly unbalanced in the discussion section. Statistical analyses did not reveal any sex differences neither for the mean score of the FMTQ nor between the two groups of childhood trauma. An ANOVA has now been conducted to assess the presence of any sex differences for the mean score of FMTQ-s. In the Type III model used in ANOVA, sums of squares are invariant with respect to the cell frequencies; hence, this type of sums of squares is often considered useful for an unbalanced model with no missing cells. Moreover, the assumption of equal variances across groups was not violated. Thus, it seems that ANOVA results can be considered reliable, despite the groups unequal sample size.

Reviewer 3 Report

Dear authors and editor,

The manuscript titled "Italian validation of the short version of the Failure to Mentalize Trauma Questionnaire (FMTQ-s) in individuals at risk for  Childhood Trauma" aimed to refine a short version of the FMTQ (FMTQ-s) and examine its psychometric properties (i.e., internal consistency, convergent and concurrent validity) in a non-clinical population of Italian adults. The goal is to create a unidimensional short version of the FMTQ that takes into account the presence of a higher-order general factor detected in the original study.

There are many minor and major issues I'd like the authors resolve.

Abstract

1-It is recommended to remove author references in the abstract of the manuscript.

2-Change the keywords.  Not found in the MeSH (Medical Subject Headings): "Childhood Trauma"; "Scalar Invariance"; "Discriminant  validity". Change to Adverse Childhood Experiences....

Introduction

3-The introduction is correct. The most important concepts of the topic are defined, as well as the objectives.

 Methods

4-It is recommended to include the code of ethics or date of acceptance.

5-It is recommended to indicate the sample size calculation. Some explanation of gender differences in the sample. These differences can be justified.

Results

6-Adequate, correct psychometric analysis is carried out.

7-What added value does it give to the study to assess populations that do not have childhood trauma?
Discusión

8-The discussion is correct. The authors summarise the findings in the results. They also compare and justify with other authors. A section on limitations is added, including possible biases.

Conclusión

9-Adequate

Referencia:

10- It is recommended to check bibliographic references.

Author Response

The manuscript titled "Italian validation of the short version of the Failure to Mentalize Trauma Questionnaire (FMTQ-s) in individuals at risk for  Childhood Trauma" aimed to refine a short version of the FMTQ (FMTQ-s) and examine its psychometric properties (i.e., internal consistency, convergent and concurrent validity) in a non-clinical population of Italian adults. The goal is to create a unidimensional short version of the FMTQ that takes into account the presence of a higher-order general factor detected in the original study.

There are many minor and major issues I'd like the authors resolve.

Authors’ answer: We would like to thank the Reviewer for all the helpful comments on the manuscript.

Abstract

1-It is recommended to remove author references in the abstract of the manuscript.

Authors’ answer: References have been removed.

2-Change the keywords.  Not found in the MeSH (Medical Subject Headings): "Childhood Trauma"; "Scalar Invariance"; "Discriminant  validity". Change to Adverse Childhood Experiences....

Authors’ answer: We have now changed the keywords by searching the appropriate terms in the MeSH.

Introduction

3-The introduction is correct. The most important concepts of the topic are defined, as well as the objectives.

Authors’ answer: Thank you.

 Methods

4-It is recommended to include the code of ethics or date of acceptance.

Authors’ answer: The protocol number of ethical approval has now been added.

5-It is recommended to indicate the sample size calculation. Some explanation of gender differences in the sample. These differences can be justified.

Authors’ answer: Concerning the minimum sample sized, we have followed the indication reported in the literature regarding factor analysis, which suggest a minimum number of individuals to be ranged between 200 and 500. We have also specified it in the manuscript. Sex difference are now better explored in the discussion section.

Results

6-Adequate, correct psychometric analysis is carried out.

Authors’ answer: Thank you.

7-What added value does it give to the study to assess populations that do not have childhood trauma?

Authors’ answer: We validated the questionnaire on a non-clinical population because adverse childhood experiences are common also in non-clinical individuals and, therefore, an instrument assessing mentalization deficits due to past traumas validated also on a sample from the general population could be helpful for epidemiological studies. We have added this consideration at the end of the Introduction.

Discusión

8-The discussion is correct. The authors summarise the findings in the results. They also compare and justify with other authors. A section on limitations is added, including possible biases.

Authors’ answer: Thank you.

Conclusión

9-Adequate

Authors’ answer: Thank you.

Referencia:

10- It is recommended to check bibliographic references.

Authors’ answer: Reference’s style was checked and corrected.

Reviewer 4 Report

Italian validation of the short version of the Failure to Mentalize Trauma Questionnaire (FMTQ-s) in individuals at risk for 3 Childhood Trauma

I read the paper very carefully, I find it interesting but I report some of my perplexities

- Why should a 29-item instrument be thought of in a short form?

- Are we sure that an instrument validated in the literature retains its diagnostic ability even when thought of in a short form?

- The items excluded from the original form were not specified in the article. Could they be specified in a different file?

- Is the mathematical and statistical correctness of item reduction through exploratory factor analysis sufficient to propose a short version?

- How come the authors proceeded to analyze data from an unbalanced sample

- Men 72 (10.2%) Women 637 (89.8%)

I will be glad to read the authors' responses before considering the work worthy of publication.

Author Response

I read the paper very carefully, I find it interesting but I report some of my perplexities

Authors’ answer: We would like to thank the Reviewer for all the helpful comments on the manuscript.

- Why should a 29-item instrument be thought of in a short form?

Authors’ answer: In general for epidemiological studies, where a large number of variables are measured through surveys, the presence of questionnaire with few items is recommended. Moreover, research has found that surveys using short questionnaire (i.e., 13-item questionnaire) compared to longer version (i.e., 25-item questionnaire) were as reliable as longer surveys and also produced higher response and completion rates than longer surveys (Kost RG, de Rosa JC. Impact of survey length and compensation on validity, reliability, and sample characteristics for ultrashort-, short-, and long-research participant perception surveys. J Clin Transl Sci 2018;2:31-7.). Therefore, our FMTQ-s can be considered a shorter version of the FMTQ in terms of number of items.

- Are we sure that an instrument validated in the literature retains its diagnostic ability even when thought of in a short form?

Authors’ answer: To assess the sensibility and specificity of this short version of the FMTQ as a diagnostic tool, future studies are needed with clinical  subjects and we have added this suggestion in the limits section. Since mentalization deficits due to past adverse childhood experience are common also in non-clinical population, the aim of our study was to provide a short and easy to administer version of the FMTQ that could be used in epidemiological studies.

- The items excluded from the original form were not specified in the article. Could they be specified in a different file?

Authors’ answer: Items that have been excluded are now specified on a separate file.

- Is the mathematical and statistical correctness of item reduction through exploratory factor analysis sufficient to propose a short version?

Authors’ answer: Although future studies are needed to assess the reliability and stability of this reduced version of the questionnaire, the approach used (i.e., to split the sample and perform the exploratory factor analysis (EFA) on one subsample and the confirmatory factor analysis on the other) is considered to be the best practice for assessing the adequacy of a factor model (Lorenzo-Seva, U. (2022). SOLOMON: A method for splitting a sample into equivalent subsamples in factor analysis. Behavior Research Methods, 54(6), 2665-2677). Through EFA we were able to assess the presence of low factor loadings (< 0.40), which indicates that those items are less useful to measure the construct of the questionnaire. Therefore, they can be removed from the model.

- How come the authors proceeded to analyze data from an unbalanced sample

- Men 72 (10.2%) Women 637 (89.8%)

Authors’ answer: Although the sample is unbalanced, we performed adequate statistical analyses that took into account the unequal proportion of men and women. An ANOVA has now been conducted to assess the presence of any sex differences for the mean score of FMTQ-s. In the Type III model used in ANOVA, sums of squares are invariant with respect to the cell frequencies; hence, this type of sums of squares is often considered useful for an unbalanced model with no missing cells. Moreover, the assumption of equal variances across groups was not violated. Thus, it seems that ANOVA results can be considered reliable, despite the groups unequal sample size.  

I will be glad to read the authors' responses before considering the work worthy of publication.

Round 2

Reviewer 3 Report

The authors have responded to the recommendations indicated. They have acknowledged the limitations of the manuscript and have improved or clarified important aspects.

Kind regards.

Reviewer 4 Report

Thank you very much for your comments. I appreciated the quality of the responses to the different points in my report.